# Global translational control by the transcriptional repressor TrcR in the filamentous cyanobacterium *Anabaena* sp. PCC 7120

Zi-Qian Wang [1,2,4 ✉], Yiling Yang [1], Ju-Yuan Zhang[1], Xiaoli Zeng[1] & Cheng-Cai Zhang [1,3 ✉]

Transcriptional and translational regulations are important mechanisms for cell adaptation to environmental conditions. In addition to house-keeping tRNAs, the genome of the filamentous cyanobacterium *Anabaena* sp. strain PCC 7120 (*Anabaena*) has a long tRNA operon (*trn* operon) consisting of 26 genes present on a megaplasmid. The *trn* operon is repressed under standard culture conditions, but is activated under translational stress in the presence of antibiotics targeting translation. Using the toxic amino acid analog β-N-methylamino-L-alanine (BMAA) as a tool, we isolated and characterized several BMAA-resistance mutants from *Anabaena*, and identified one gene of unknown function, *all0854*, named as *trcR*, encoding a transcription factor belonging to the ribbon-helix-helix (RHH) family. We provide evidence that TrcR represses the expression of the *trn* operon and is thus the missing link between the *trn* operon and translational stress response. TrcR represses the expression of several other genes involved in translational control, and is required for maintaining translational fidelity. TrcR, as well as its binding sites, are highly conserved in cyanobacteria, and its functions represent an important mechanism for the coupling of the transcriptional and translational regulations in cyanobacteria.

[1] State Key Laboratory of Freshwater Ecology and Biotechnology and Key Laboratory of Algal Biology, Institute of Hydrobiology, Chinese Academy of Sciences, Wuhan, Hubei 430072, People's Republic of China. [2] University of Chinese Academy of Sciences, Beijing 100049, People's Republic of China. [3] Institute AMU-WUT, Aix-Marseille Université and Wuhan University of Technology, Wuhan, Hubei, People's Republic of China. [4] Present address: State Key Laboratory of Microbial Resources, Institute of Microbiology, Chinese Academy of Sciences, Beijing, China. ✉email: wangziqian@im.ac.cn; cczhang@ihb.ac.cn

Transcriptional and translational regulations are critical mechanisms for an organism to adapt to environmental changes. Under certain stress conditions, for example, cells tend to reduce global protein synthesis while upregulating the expression of genes involved in stress resistance[1]. Compared to transcriptional control, regulation at the translational level has more direct and rapid impact on protein amounts and activities. Translational control is a complex process, which may occur at multiple levels, such as the regulation of aminoacyl-tRNA synthetase (aaRS) activity, changes of tRNA pool, ribosome heterogeneity, tRNA modification, translational fidelity control, and selective mRNA translation, etc.[1–4].

Cyanobacteria are the earliest known microorganisms on Earth to produce oxygen through photosynthesis[5]. Like other organisms, cyanobacteria coordinate both transcriptional and translational processes for better adaptation to environmental changes and fitness maintenance. However, translational regulation has been poorly studied in cyanobacteria. Recently, a few studies explored the contribution of post translational modification on environmental adaptability[6,7]. Ignacio Luque et al. found that in the cyanobacterium Anabaena sp. strain PCC 7120 (Anabaena hereafter), a tRNA operon (trn) was activated when cells encountering translational stress[8]. Two types of tRNA genes were found in Anabaena. One of these types contains 48 tRNA genes that are scattered on the chromosome and transcribed under standard laboratory conditions. These tRNA genes encode housekeeping tRNAs[8,9]. The other type includes 26 tRNA genes that constitute the trn operon, which is on the δ plasmid and is silent under normal laboratory culture conditions, but activated when cells encounter translational stress[8]. The transcription of the trn array was induced by addition of antibiotics targeting translation, and increased trn expression favored survival of Anabaena under translational stress induced by antibiotics[8]. This study proved that a large tRNA array (trn operon), normally silenced, could be activated and participate in the process of translational regulation. However, the underlying mechanism for the regulation of this large trn operon remained unknown[8].

Amino acids analogs, by disturbing the translational process, have been extensively used for probing translational responses in various organisms[10–12]. One of such analogs, β-N-methylamino-L-alanine (BMAA), is a non-protein amino acid. BMAA inhibits both the proofreading activity and the alanine aminoacylation activity of the human alanyl-tRNA synthetase, and is also a substrate of the human alanyl-tRNA synthetase by forming BMAA-tRNA$^{Ala}$ [13]. In addition, some studies have demonstrated that BMAA can be incorporated into polypeptides during protein synthesis[14–17]. Therefore, BMAA is able to cause translational stress in cells through multiple mechanisms. Previously, we provided evidence that BMAA, as an amino acid analog toxic to cyanobacteria, could be used as a valuable tool for the studies of amino acid transport as well as translational control in cyanobacteria such as Anabaena[18,19]. Anabaena is a filamentous and multicellular cyanobacterium. In addition to using combined nitrogen such as nitrate and ammonium as nitrogen sources, it can also fix atmospheric nitrogen through heterocysts, whose differentiation is induced under the condition of combined-nitrogen deficiency[20]. Previously, using Anabaena as a model organism and BMAA as a molecular tool, through screening of BMAA-resistance mutants and genetic analysis, we explored the toxic mechanisms of BMAA to cyanobacteria and the translational stress response when cells are challenged by BMAA[18,19]. By this method, we found that BMAA is imported into Anabaena mainly through N-I and N-II amino acid transport systems, and that the N(6)-threonylcarbamoyl adenosine (t$^6$A) modification of tRNA plays an important role in translational regulation[18,19]. In addition to BMAA, antibiotics targeting different steps of the translational process are also used for applying the translational stress by inhibiting translation. For example, kasugamycin (Ksg) blocks translation initiation by targeting the 30 S ribosomal subunit[21], while chloramphenicol (Cm) inhibits translation elongation by binding to the 50 S ribosomal subunit[22].

In this study, we report the analysis of several mutations conferring BMAA-resistance, and occurring in all0854, which we annotated as trcR (Translational Control Regulator). TrcR is a transcriptional repressor with a global impact on the expression of genes involved in translational processes. Furthermore, we show that TrcR is a repressor for the transcription of the large trn operon, thus providing a regulation mechanism on the expression of this trn array. The role of TrcR in translational control constitutes a new mechanism for the coupling of transcriptional and translational regulations reported in cyanobacteria.

## Results

### trcR (all0854), a gene conferring BMAA sensitivity in Anabaena.
Previously, we showed that BMAA-resistance mutants allowed us to study the mechanism of BMAA toxicity and translational control in cyanobacteria[18,19]. Among the 20 BMAA-resistance mutants obtained, 17 (M1-M17) of them have been described in the previous studies[18,19]. Here, we focused on M18, M19, and M20, the remaining three BMAA-resistance mutants that could still grow, despite poorly, in the presence of 50 μM BMAA, while the WT growth was completely inhibited under the same conditions (Supplementary Fig. 1, Supplementary Table 1).

Whole-genome resequencing of M20 revealed that a single transition mutation of T131C occurred in trcR (all0854), resulting in a replacement of Leu at position 44 by a Pro residue in the corresponding protein (Supplementary Table 1 and Supplementary Data 1). Our previous studies on the other BMAA-resistance mutants have identified three genes, alr4167, all1284 and alr2300, which play roles in amino acid uptake (alr4167 and all1284) or t$^6$A modification of tRNA (alr2300) (Supplementary Table 1)[18,19]. Therefore, for the other two mutants M18 and M19, we checked whether any mutation occurred in trcR, alr4167, all1284 or alr2300 by PCR coupled with sequencing. The results showed that both also had mutations in trcR, with M19 having the same mutation as that in M20, and M18 having a mutation of C215A that resulted in a replacement of Ala72 by an Asp residue in TrcR (Supplementary Table 1).

To further verify that the mutation in trcR is responsible for BMAA resistance, one deletion mutant ΔtrcR was created. We found that ΔtrcR was resistant to BMAA in contrast to the WT (Fig. 1). Furthermore, complementation of M20 and ΔtrcR with trcR (M20-CtrcR or C-trcR) fully restored their BMAA sensitivity to the WT level (Fig. 1). These results confirmed that the mutation of trcR is responsible for BMAA resistance of Anabaena.

### TrcR is an autoregulated transcriptional repressor.
TrcR was annotated as an unknown protein in the data banks. We made a sequence alignment and found that TrcR contains a region with a ribbon-helix-helix (RHH) domain (Supplementary Fig. 2). Proteins of the RHH family may bind DNA in a sequence-specific manner, and thus function as transcription factors[23–25]. To test if TrcR is a transcription factor, we tested whether it binds to its own promoter, since many transcription factors are autoregulated. Sequence alignment of the promoter regions of trcR and its homologous genes from other cyanobacteria revealed the presence of a highly conserved DNA motif (5'-ATACTACA CTTGTATTAC-3') (Fig. 2a). Five DNA fragments containing or nearby the promoter region of trcR were then selected as the candidate substrates of TrcR (Fig. 2b). The EMSA results showed

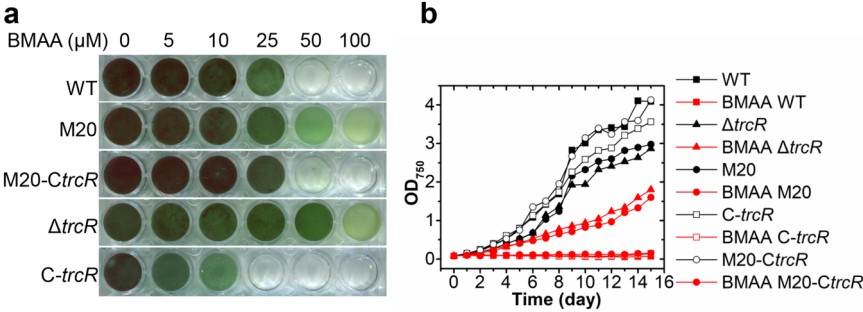

**Fig. 1 BMAA sensitivity test in the WT, M20, Δ*trcR*, M20-C*trcR* and C-*trcR*. a** Cell growth in 24-well plates containing indicated concentrations of BMAA in BG11 medium. **b** Growth curves of the same strains with 25 µM BMAA (red) or without BMAA (black). Data represents the mean values of two independent experiments.

that DNA3 and DNA4, two DNA fragments containing an overlapping region, showed retarded migration on the gel due to TrcR binding (Fig. 2c). Further experiment showed that competition for TrcR binding could be observed between unlabeled DNA3 and DNA4 and the respective FAM-labeled fragments. The competition of unlabeled DNA4 with FAM-labeled DNA3, or of unlabeled DNA3 with FAM-labeled DNA4, also occurred ; in contrast, no such competition happened with DNA1 (Fig. 2, panels d and e; Supplementary Fig. 3), demonstrating that TrcR specifically binds to DNA3 and DNA4.

To determine the binding site of TrcR, truncated forms of DNA3 and DNA4 were tested for their TrcR binding activity by EMSA (Supplementary Fig. 3c, d). The results confirmed that a region of 25 bp from −50 to −25 (relative to the translational start site of *trcR*) is essential for TrcR binding (Supplementary Fig. 3). Footprinting assay revealed that a DNA fragment from -53 to -28 was protected by TrcR from DNase I digestion (Fig. 2f). As expected, both regions obtained via EMSA and Footprinting assays contain the motif (5'-ATACTACACTTGTATTAC-3') conserved in front of *trcR* and its homologs in other cyanobacteria (Fig. 2 and Supplementary Fig. 3).

Based on the RNA-Seq results[26], the −10 box region 'TACACT' and transcription start site (TSS) of *trcR* overlaps with TrcR binding site, (Fig. 2b), suggesting that TrcR could have an autorepression function. To confirm this hypothesis, a plasmid expressing CFP under the control of the *trcR* promoter (p$_{trcR}$CFP) was transformed into WT, M20 or Δ*trcR*. Strong CFP fluorescence was observed in the two mutants (M20::p$_{trcR}$CFP and Δ*trcR*::p$_{trcR}$CFP), while little fluorescence was detected in the WT (WT::p$_{trcR}$CFP) (Fig. 3a). This result suggested that TrcR acted as an autorepressor. To further test this hypothesis, we expressed TrcR in the WT or Δ*trcR* using a plasmid p$_{CT}$TrcRp$_{trcR}$CFP in which *trcR* was controlled by an inducible CT promoter, allowing protein expression only in the presence of inducers (Cu$^{2+}$ and theophylline), and the p$_{trcR}$CFP fusion was also present as a reporter on the same plasmid[27]. As expected, in the absence of inducers, the *trcR* mutant (Δ*trcR*::p$_{CT}$TrcR-p$_{trcR}$CFP) exhibited stronger CFP fluorescence than that of WT (WT::p$_{CT}$TrcR-p$_{trcR}$CFP). However, when inducers were supplied in the medium, the fluorescence disappeared in the *trcR* mutant (Δ*trcR*::p$_{CT}$TrcR-p$_{trcR}$CFP), while no fluorescent change was observed in the same mutant expressing the transcriptional fusion alone (Δ*trcR*::p$_{trcR}$CFP) (Fig. 3b, c), demonstrating that the production of TrcR in cells led to repression of the *trcR* promoter. Our results, all together, proved that TrcR binds to its own promoter and acts as a transcriptional repressor.

By aligning TrcR with other RHH family proteins, we found that the mutation in M20 resulted in a change from a Leu residue at position 44 to a Pro residue, and this Leu44 is conserved at the helix B in TrcR (Supplementary Fig. 2). This observation suggests

that Leu44 is essential for the DNA-binding activity of TrcR. Therefore, we tested the DNA-binding activity of a corresponding mutant form of TrcR (TrcR-L44P, TrcR with Leu[28] to Pro[28] mutation), and the results indicated that TrcR-L44P lost the binding activity to the promoter of *trcR* (Fig. 4a–c). These results, together with those obtained with other target-binding sites as described below, demonstrated the binding specificity of TrcR.

**TrcR regulates transcription of genes involved in translation.** Next, to identify genes of the TrcR regulon, we compared the transcriptome of WT and Δ*trcR*. RNA-seq data showed that 266 genes were differentially expressed in Δ*trcR*, with at least 2-fold changes when compared to WT. Among the 266 genes, 203 genes were upregulated and 63 genes downregulated in Δ*trcR* (Supplementary Fig. 4, Supplementary Data 2 and Data 3). We narrowed down the candidate genes to 18 by choosing those with more than 8-fold changes, and all of them were upregulated in Δ*trcR* (Table 1), consistent with the idea that TrcR functions mainly as a repressor. EMSA were performed to test the interaction between TrcR and the promoters of the candidate genes, except for the promoter between *asr0855-all0854* (Fig. 2b), which was already confirmed (Fig. 2). Note that only the promoter sequence of the first upstream gene was selected for those consecutive genes that may constitute an operon, i.e. *alr3301* promoter for *alr3301-3303*, *all0263* promoter for *all0261-0263*, *alr0739* promoter for *alr0739-0740* and *alr1537* promoter for *alr1537-1540*. The EMSA results showed that except for the promoter of *all5040* that lacks a TrcR binding site, all the tested DNA fragments exhibited band shift in the presence of TrcR (Fig. 4a). Among those target genes, *all3526*, *alr3303* and *alr8077* are related to translation. *all3526* (*rtcB*) encodes a widely distributed RNA ligase RtcB, which is involved in tRNA intron splicing in eukaryotes and archaea[29], and the repair of cleaved 16 S rRNA and tRNAs in bacteria[30,31]. *alr8077* (*rsgA*) encodes the ribosome assembly factor RsgA that is involved in the late stages of 30 S subunit maturation[32,33]. *alr3303*, which is cotranscribed with *alr3301* and *alr3302* (Supplementary Fig. 5), encodes a ribosome modification protein RimK, an ATP-dependent glutamate ligase that adds glutamate residues to the C-terminus of the ribosomal protein S6[34]. The post-translational modification of S6 has been shown to be important for translational control and environmental adaptation in cells[35–37].

To further confirm the transcriptome data, qRT-PCR was performed to check the transcript levels of *all3526*, *alr8077*, and *alr3303* in the WT, Δ*trcR* and C-*trcR* strains (Fig. 4d). The results revealed that the transcription levels of these genes were significantly upregulated when TrcR was absent in the cells. The observed regulation was further validated by using CFP as a reporter (Supplementary Fig. 6). A transcriptional fusion of

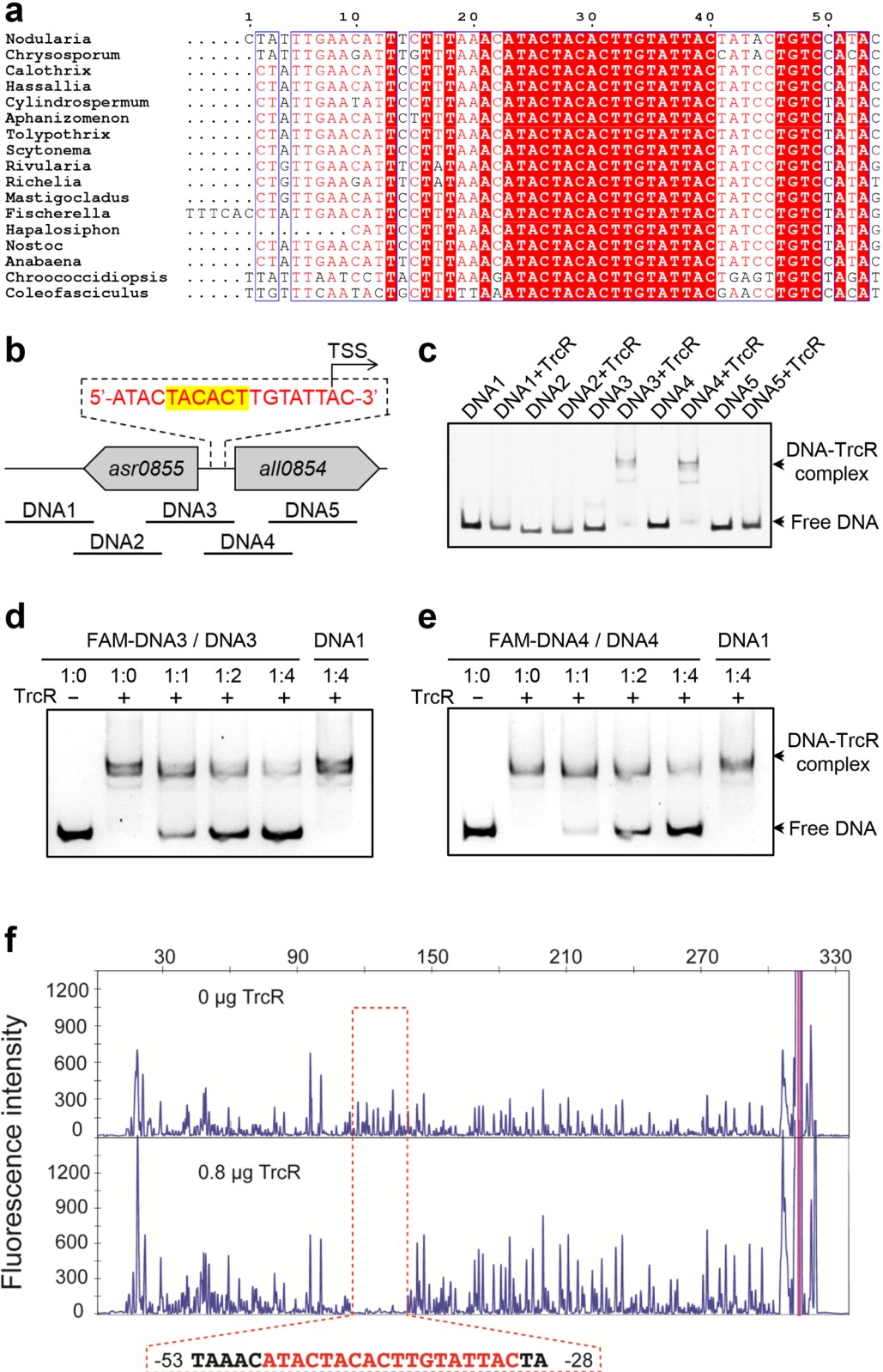

promoter region of *all3526*, *alr3301* or *alr8077* was made and transferred, respectively, into the WT or the Δ*trcR* mutant. As expected, all the CFP fusions showed stronger CFP fluorescence in Δ*trcR* than that in the WT (Supplementary Fig. 6). Together, these results demonstrate that TrcR is a transcriptional repressor that prevents transcription of *all3526*, *alr3301* and *alr8077*, genes related to translational process.

Further footprinting assays using the promoter sequences of *alr3301*, *all3526*, *alr8077*, *all8564* and *alr1537* (Supplementary Fig. 7) revealed regions protected by TrcR. By aligning all the determined binding sites of TrcR, we determined a consensus binding sequence of 8 bp in length for TrcR (Fig. 4e). The position of this motif overlaps with or is located nearby the -10 boxes of the corresponding promoters, consistent with TrcR being a repressor (Fig. 4f).

**Fig. 2 TrcR binds to a conserved DNA motif at its own promoter region. a** Sequence alignment of the promoter regions of *trcR* and its homologous genes from different cyanobacteria. Nodularia: *Nodularia spumigena* CCY9414; Chrysosporum: *Chrysosporum ovalisporum*; Calothrix: *Calothrix* sp. PCC 7507; Hassallia: *Hassallia byssoidea* VB512170; Cylindrospermum: *Cylindrospermum stagnale* PCC 7417; Aphanizomenon: *Aphanizomenon flos-aquae* 2012/KM1/D3; Tolypothrix: *Tolypothrix* sp. PCC 7601; Scytonema: *Scytonema tolypothrichoides* VB-61278; Rivularia: *Rivularia* sp. PCC 7116; Richelia: *Richelia intracellularis*; Mastigocladus: *Mastigocladus laminosus* UU774; Fischerella: *Fischerella* sp. JSC-11; Hapalosiphon: *Hapalosiphon* sp. MRB220; Nostoc: *Nostoc punctiforme* PCC 73102; Anabaena: *Anabaena* sp. PCC 7120; Chroococcidiopsis: *Chroococcidiopsis thermalis* PCC 7203; Coleofasciculus: *Coleofasciculus chthonoplastes* PCC 7420. **b** Schematic Illustration of the 5 DNA fragments selected for EMSA. −10 box (yellow background) and transcription start site (TSS) of *trcR* based on the RNA-Seq data from Mitschke et al.[26] are shown in the conserved motif identified in **a**. **c** EMSA performed with DNA1 to DNA 5 in the presence or absence of TrcR. **d, e** EMSA competition with unlabeled DNA. **d** unlabeled DNA3 competes the binding of TrcR with FAM-labeled DNA3. **e** unlabeled DNA4 competes the binding of TrcR with FAM-labeled DNA4. **f** TrcR binding region determined by DNase I footprinting. The TrcR protected region is indicated by a red box with sequences shown below (0.8 μg TrcR, lower panel). As a negative control, the corresponding region without TrcR addition was also shown (upper panel). Letters in red indicates the conserved motif identified in (**a**).

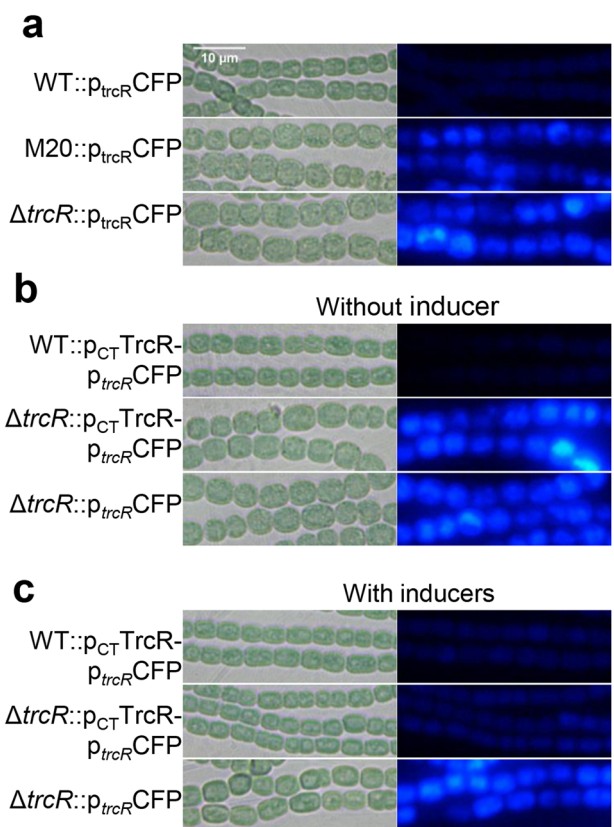

**Fig. 3 TrcR is an autorepressor as shown with microscopic images of CFP reporter fluorescence. a** A plasmid bearing a CFP reporter gene under the control of the *trcR* promoter ($p_{trcR}$CFP) in WT, M20 and Δ*trcR* backgrounds. **b** WT or Δ*trcR* bearing a replicative plasmid with the expression of TrcR driven by an inducible system (the CT promoter, $P_{CT}$), together with $p_{trcR}$CFP as in (**a**). No inducers were added, thus no induction of TrcR from the plasmid. Δ*trcR*::$p_{trcR}$CFP as in A was used as a control. **c** Same experiments performed as in (**b**) but with addition of inducers (0.6 μM $Cu^{2+}$ + 2 mM Tp) in the growth medium for the expression of *trcR* from the inducible promoter $P_{CT}$, and CFP expressed from the promoter of *trcR* ($p_{trcR}$CFP).

**TrcR is the repressor for the silenced *trn* gene array.** The analysis of transcriptome data did not include long non-coding RNAs. Recently, a *trn* operon consisting of 26 tRNA genes was found to be activated under translational stress triggered by treatment with the antibiotic Cm[8]. In addition, the same study found also 1496 genes downregulated and 1750 genes upregulated under the same stress conditions[8]. Interestingly, among those upregulated genes, *trcR*, *all0768*, *all3526*, *all8564*, *alr0739*, *alr0740*, *alr1537*, *alr1538*, *alr1539*, *alr1554*, *alr8077*, *asr0855* and *alr0857*

were also upregulated by more than 8 folds in the Δ*trcR* strain in this study (Table 1). Therefore, TrcR could be the repressor in *Anabaena* responsible for silencing the expression of the *trn* operon under standard culture conditions. We first checked by qRT-PCR the expression of *trn* operon in the WT, the Δ*trcR* strain and the complemented strain (Fig. 5). While the expression of this operon remained low in the WT and the complemented strain, a highly activated expression was detected in the deletion mutant (Fig. 5a). As controls, the expression levels of the three housekeeping tRNA genes, *allrt06*, *allrt16* and *allrt02*, remained relatively constant (Fig. 5b). Consistent with these results, a sequence 5'-TGTAGTAT-3', similar to the consensus-binding site of TrcR overlaps with the -10 box (5'-TAGTAT-3') of the promoter of the *trn* operon (Fig. 5c). By footpringing experiment, this putative binding site of TrcR was confirmed since TrcR could efficiently protect it against DNase I digestion (Fig. 4f and Supplementary Fig. 7).

To further confirm that the *trn* array is repressed by TrcR, several strategies were used. In vitro, EMSA experiments indicated that TrcR, but not TrcR-L44P, could bind to the promoter region of the *trn* operon (Fig. 4a and c). In vivo, using CFP as a reporter, the fluorescence intensity of CFP driven by the promoter of the *trn* operon, was strongly upregulated in the Δ*trcR* strain, while it was hardly visible in the WT under the same experimental conditions (Supplementary Fig. 6).

**Translational stress relieves the repression of TrcR on its regulatory genes.** Previously, it was reported that the *trn* operon was induced by antibiotics targeting ribosome, and deletion of *trn* reduced the resistance of *Anabaena* to this type of antibiotics[8]. To test whether antibiotic treatment could also cause the expression of genes under the control of TrcR, the transcription levels of *trcR*, *all3526*, *alr3303*, *alr0877* and *trn* were visualized via transcriptionally fused CFP gene under each specific promoter in the WT strain upon antibiotic treatment (Fig. 6a and Supplementary Fig. 8). Antibiotics disrupting translational processes were chosen, including Cm, Ksg, and streptomycin (Sp) that inhibits translational elongation by targeting the ribosomal 30 S subunit[38], and streptomycin (Sm) that interferes both the selection of aminoacyl-tRNA and the translational proof-reading activity by also targeting 30 S subunit[38]. Penicillin G (PenG) targeting cell wall biosynthesis was used as a control. All antibiotics were applied at sub-lethal concentrations (Supplementary Fig. 9). Our results showed that Cm, Sp and Sm induced CFP fluorescence in all strains, while PenG and Ksg did not (Fig. 6a and Supplementary Fig. 8). qRT-PCR result confirmed that the transcripts of *trcR*, *all3526*, *alr3303*, *alr0877* and *trn* were significantly increased with time under Cm-induced stress (Supplementary Fig. 10).

Unexpectedly, Ksg, as an antibiotic acting on translational initiation, did not cause a derepression of TrcR. To understand this point, we tested the sensitivity of the WT, Δ*trcR* and C-*trcR*

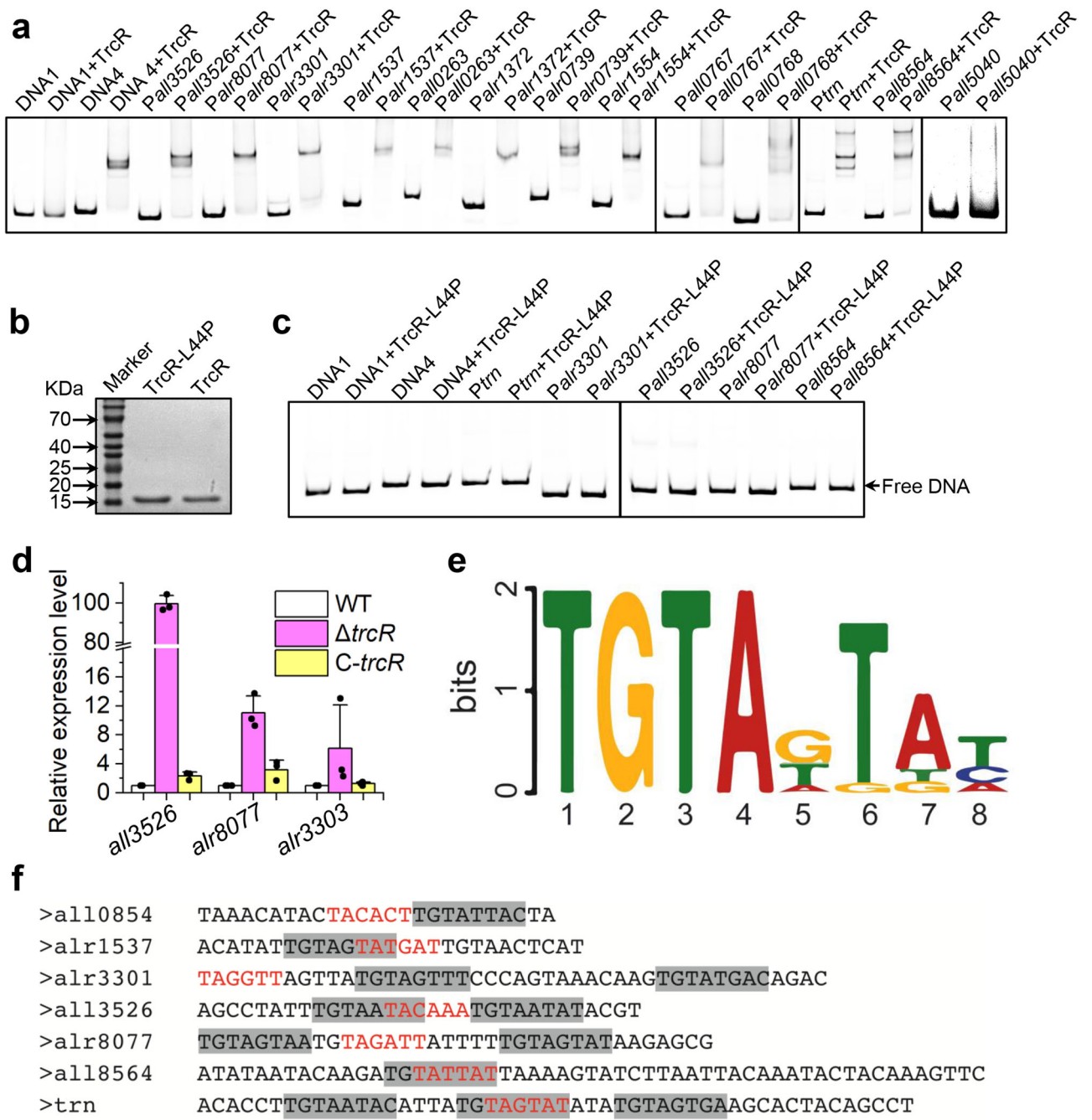

**Fig. 4 Regulation of several genes under the control of TrcR. a** EMSA showing the binding of TrcR to the promoter regions of *trn* and other potential target genes (with more than 8-folds upregulation in Δ*trcR*). **b** Purification and quantification of TrcR and TrcR-L44P (TrcR bearing a mutation with replacement of L44 by P) used for EMSA by SDS-Polyacrylamide Gel Electrophoresis following Coomassie blue staining. **c** Binding assay of TrcR-L44P to the promoters of *trn, alr3301, all3526, alr8077* and *all8564* tested by EMSA. No binding to any promoter was detected. **d** Transcription levels of *all3526, alr8077* and *alr3303* quantified by qRT-PCR in WT, Δ*trcR* and C-*trcR*. n = 3 biologically independent samples. The experiment was repeated three times for each sample. Data shown are the mean values ± S.D. **e** The consensus binding motif of TrcR based on the DNA region protected by TrcR in footprint experiments. Analysis was performed by using MEME on website (https://meme-suite.org/meme/tools/meme). **f** Relative positions of the identified TrcR binding motif (shaded background) and the −10 boxes (red letters) are shown in the promoter regions of *trcR, alr1537, alr3301, all3526, alr3077, all8564* and *trn*. The positions of −10 boxes were deduced based on the RNA-Seq data from Mitschke et al.[26].

to Cm, Ksg, Sp, Sm and PenG. All of them had a similar sensitivity to Cm, Sp, Sm and PenG, but Δ*trcR* gained resistance to Ksg (Supplementary Fig. 9). To explore whether the higher resistance of Δ*trcR* to Ksg and BMAA is the consequence of gene overexpression, we deleted *all3526, alr8077, trn* operon and *alr3301-alr3303* operon from Δ*trcR*, respectively, to create the double mutants Δ*trcR*Δ*all3526*, Δ*trcR*Δ*alr8077*, Δ*trcR*Δ*trn* and

Δ*trcR*Δ*alr3301-03*. We found that deletion of these genes from Δ*trcR* had no influence on its higher resistance to Ksg and BMAA (Supplementary Figs. 11 and 12). So, further investigation is required to understand the resistance of Δ*trcR* to Ksg and BMAA.

Another paradox is that Cm, and to a lesser extent BMAA (Supplementary Figs. 10 and 13), induced the expression of *trcR*, which normally would lead to stronger repressive effects for the

genes of the TrcR regulon; yet, under the same conditions, genes repressed by TrcR were induced. To understand the mechanism of antibiotic-induced derepression from TrcR, we checked the TrcR level in *Anabaena* exposed to Cm, Ksg and PenG by Western Blot. The results showed that the amount of TrcR significantly decreased with time when 5 µg/mL Cm was added

**Table 1 List of genes with more than 8-fold increase in the transcriptional levels in Δ*trcR*.**

| Gene ID | Fold change | Function |
|---|---|---|
| *all0261* | 4.052915 | sugar transport system permease protein |
| *all0262* | 10.63186 | threonine dehydratase |
| *all0263* | 23.84687 | AAA family ATPase |
| *all0767* | 22.02014 | endoribonuclease L-PSP |
| *all0768* | 322.6987 | GNAT family N-acetyltransferase |
| *all3526* | 77.33209 | RtcB family protein; RNA-splicing ligase RtcB |
| *all5040* | 13.73592 | DUF1194 domain-containing protein |
| *all8564* | 17.32671 | HNH endonuclease; Restriction endonuclease |
| *alr0739* | 71.56269 | hypothetical protein |
| *alr0740* | 42.48551 | slipin family protein; membrane protease subunit |
| *alr1372* | 12.89042 | TIGR02452 family protein |
| *alr1537* | 50.08121 | GNAT family N-acetyltransferase |
| *alr1538* | 38.23092 | DMT family transporter |
| *alr1539* | 23.77193 | cupin domain-containing protein |
| *alr1540* | 6.225956 | N-acetylmuramoyl-L-alanine amidase |
| *alr1554* | 112.6749 | ABC transporter ATP-binding protein |
| *alr3301* | 8.717753 | hypothetical protein |
| *alr3302* | 6.700311 | hypothetical protein |
| *alr3303* | 3.89125 | Ribosomal protein S6 modification protein RimK |
| *alr8077* | 10.00235 | ribosome-associated GTPase |
| *asr0855* | 9.186091 | hypothetical protein |
| *alr0856* | 7.226703 | HNH endonuclease |
| *alr0857* | 8.479061 | hypothetical protein |

Neighbor genes that may be transcribed from the same operon are also listed.

into the medium. Though 5 µg/mL of Ksg and 1.5 µg/mL of PenG also caused a decrease of TrcR, it only lasted for the first 3 h and then was kept at a stable level (Fig. 6b). This observation led to the conclusion that when *Anabaena* is under translational stress of Cm, the protein level of TrcR will decrease despite the enhanced transcription of *trcR* under similar conditions. The decreased amount of TrcR resulted in the expression of genes that were normally repressed (Fig. 6c). These results also suggest the existence of a translational or a posttranslational regulation of TrcR for the control of its protein level.

**Deletion of *trcR* lowers translational fidelity in *Anabaena*.** Our previous study showed that BMAA affected translational fidelity in *Anabaena*[19]. Since the inactivation of *trcR* had a strong impact on the expression of a number of genes involved in translation, we tested the effect of such a misregulation by measuring translational fidelity using a series of plasmids carrying the *lacZ* gene reporter or its derivatives as previously described[19]. Three categories of the *lacZ* derivatives were used: with the initiation codon AUG of *lacZ* replaced by AUA, AUC or CUG; with *lacZ* that has +1 or -1 frameshift mutation at the eighth codon, or with nonsense mutation introduced into *lacZ* by changing its eighth codon to UAA, UAG or UGA. We found that compared with the WT, the relative β-galactosidase activity was significantly higher in Δ*trcR* bearing all three types of LacZ mutant variants (Fig. 7). This result indicated that in the absence of TrcR, the translational machinery was more likely to misread the codons or misincorporate an amino acid residue. Therefore, deletion of *trcR* leads to decreased translational fidelity, including leaky scanning of initiation codons, frameshift mutation or reading through of stop codons.

**TrcR regulates the expression of *alr1537-alr1540* responsible for BMAA export and resistance.** Till now, we know that TrcR is a transcriptional repressor involved in translational control, but it is still unclear why TrcR deletion caused increased resistance to BMAA in *Anabaena*. From the transcriptome data, we found that *alr1538* encoding a DMT family (Drug/Metabolite Transporter

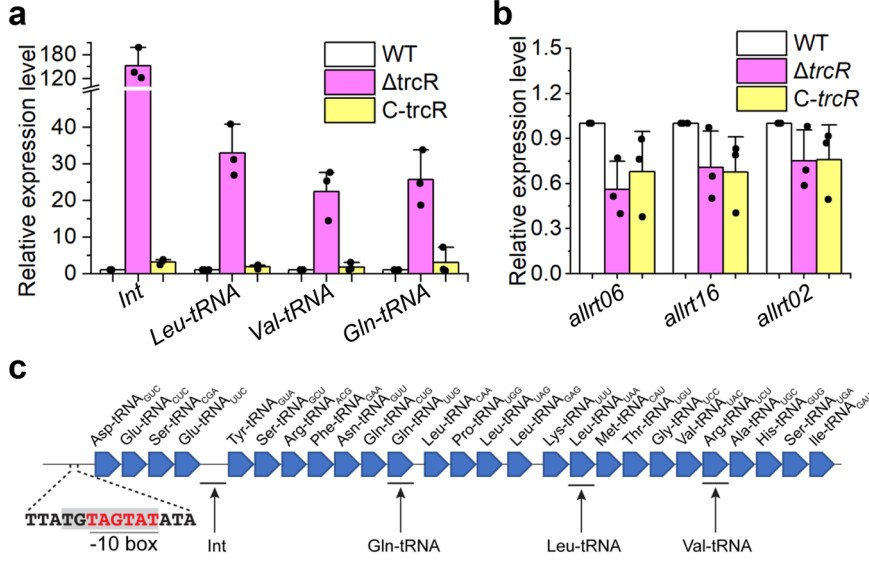

**Fig. 5 TrcR represses the *trn* operon expression. a, b** Transcription levels of the selected genes (*trn*, *allrt06*, *allrt16* and *allrt02*) quantified by qRT-PCR in WT, Δ*trcR* and C-*trcR*. *n* = 3 biologically independent samples. The experiment was repeated three times for each sample. Data shown are the mean values ± S.D. **c** Schematic illustration of the *trn* operon. The cognate amino acid and the anticodon are used for the naming of tRNA. For example, Asp-tRNA_GUC represents the tRNA whose cognate amino acid is Asp and the anticodon of which is GUC. The name and sites of fragments amplified by qRT-PCR in (**a**) were indicated below. The positions of −10 box (red letters) and a sequence TGTAGTAT (shaded background), similar to the consensus-binding site of TrcR are also indicated.

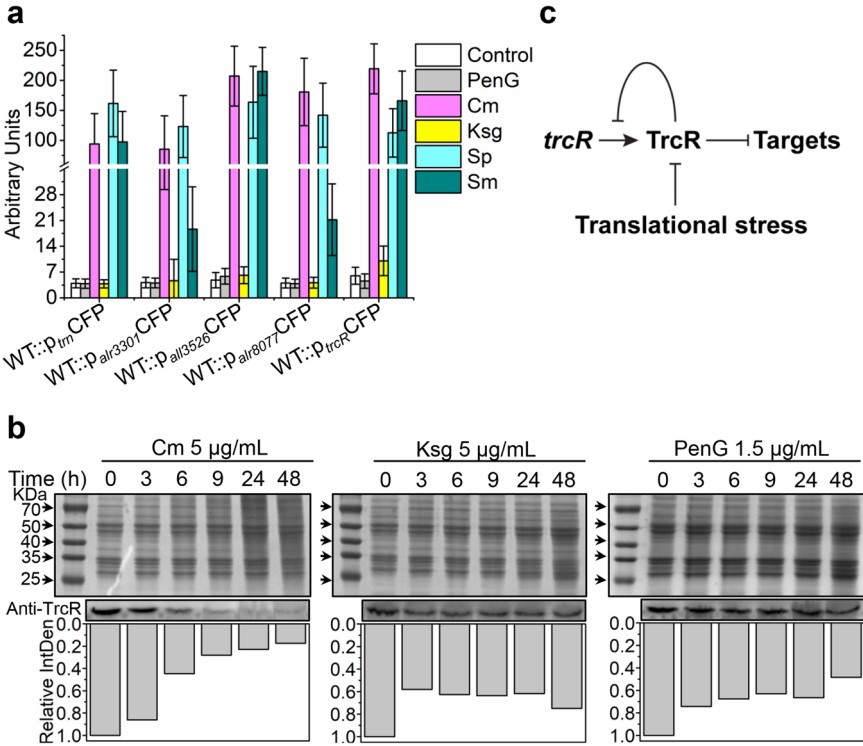

**Fig. 6 TrcR represses gene expression and its protein level is downregulated by translational stress. a** Gene expression examined through the level of CFP reporter fluorescence by transcriptional fusion. The relative fluorescence intensity of WT::$p_{trn}$CFP, WT::$p_{alr3301}$CFP, WT::$p_{all3526}$CFP, WT::$p_{alr8077}$CFP and WT::$p_{trcR}$CFP was quantified from microscopic images shown in Fig. S8 using ImageJ. All cells were incubated in BG11 containing one of the antibiotics (PenG 1.5 μg/mL, Cm 5 μg/mL, Ksg 5 μg/mL, Sp 0.3 μg/mL, Sm 0.075 μg/mL) for 48 h before imaging. **b** Effects of antibiotics (Cm, Ksg and PenG) on TrcR protein levels tested by Western blot. Total proteins of *Anabaena* collected at indicated time points after antibiotic treatment were separated by SDS-PAGE (upper panel). Western blot was carried out with antibody against TrcR (anti-TrcR, middle panel). Quantification of the relative integrated density (IntDen) of the TrcR band analyzed by ImageJ was shown below. **c** Illustration for the regulation strategy of TrcR under translational stress.

Superfamily) transporter was upregulated more than 8 folds in Δ*trcR* (Table 1). Proteins of this family are involved in the export of metabolites such as amino acids and their precursors, nucleosides and purine bases[39]. First, we used RT-PCR experiments to determine that *alr1537*, *alr1538*, *alr1539* and *alr1540* were cotranscribed, thus constituted an operon (Supplementary Fig. 14). Next, we investigated the regulation of TrcR on this operon via EMSA and qRT-PCR. Our EMSA result demonstrated that TrcR bound to the promoter region of *alr1537*, in contrast to the mutant variant TrcR-L44P that did not (Fig. 8a). The transcription level of *alr1537-alr1540* is significantly higher in Δ*trcR* compared to the WT and C-*trcR*, as revealed by qRT-PCR (Fig. 8b).

To confirm that the increased BMAA resistance of Δ*trcR* is caused by the upregulated expression of *alr1537-alr1540*, these four genes were deleted, respectively, in Δ*trcR*, resulting in the following double mutants: Δ*trcR*Δ*alr1537*, Δ*trcR*Δ*alr1538*, Δ*trcR*Δ*alr1539* and Δ*trcR*Δ*alr1540*. We also constructed a quintuple mutant Δ*trcR*Δ*alr1537-40* by deleting the whole *alr1537-alr1540* operon in Δ*trcR*. Our results showed that while Δ*trcR* kept growing in the presence of 100 μM BMAA, deletion of *alr1538* or the entire operon in Δ*trcR* restored BMAA sensitivity, with growth inhibited at 25 μM of BMAA, similarly as the WT and C-*trcR* (Fig. 8c). However, deletion of *alr1537*, *alr1539* or *alr1540* in Δ*trcR* only partially restored BMAA sensitivity, with no cell growth observed at 100 μM of BMAA and weaker cell growth observed at 50 μM of BMAA as compared to Δ*trcR* (Fig. 8c). To see if the increased Ksg resistance of Δ*trcR* is also related to *alr1537-alr1540* operon, the growth of Δ*trcR*Δ*alr1537-40* was compared to that of Δ*trcR* under different Ksg

concentrations. The result showed that Δ*trcR*Δ*alr1537-40* still exhibited increased Ksg resistance comparing to the WT, similar as that of Δ*trcR* itself (Supplementary Fig. 15).

To confirm that the efflux of BMAA was responsible for BMAA resistance in Δ*trcR*, we verified both BMAA import and export abilities of the mutants. Exogenous BMAA was added into the cultures at a final concentration of 50 μM, then cells were harvested at indicated time points for BMAA detection. Our results showed that 15 min incubation resulted in an accumulation of similar amounts of BMAA in Δ*trcR* and WT, while BMAA was undetectable in the negative control strain Δ*natA*Δ*bgtA*, a double mutant of amino acids transporters required for BMAA uptake (Fig. 8d)[18]. This result indicates that deletion of *trcR* has no influence on BMAA uptake in *Anabaena*. To further test the export of BMAA, cell samples with 15-min BMAA pre-incubation were prepared at 10, 30 and 60 min after removal of BMAA from the culture medium. Compared to WT and C-*trcR*, Δ*trcR* cells exhibited significantly decreased level of intracellular BMAA over time, concomitantly, the corresponding supernatant had increased amount of BMAA detected (Fig. 8e, f), indicating enhanced ability of BMAA export in the absence of TrcR. However, deletion of *alr1538* or *alr1537-1540* operon in Δ*trcR* fully restored both the amounts of intracellular and secreted BMAA to the WT levels (Fig. 8e, f), strongly suggesting that the protein product of *alr1538* participates in BMAA export in the absence of TrcR. Overall, our results demonstrate that *alr1537-1540* operon contribute to increased BMAA export, with *alr1538* playing a dominant role whose product could function as an efflux pump for BMAA. These results provide a rational for the higher BMAA resistance in the *trcR* deletion mutant.

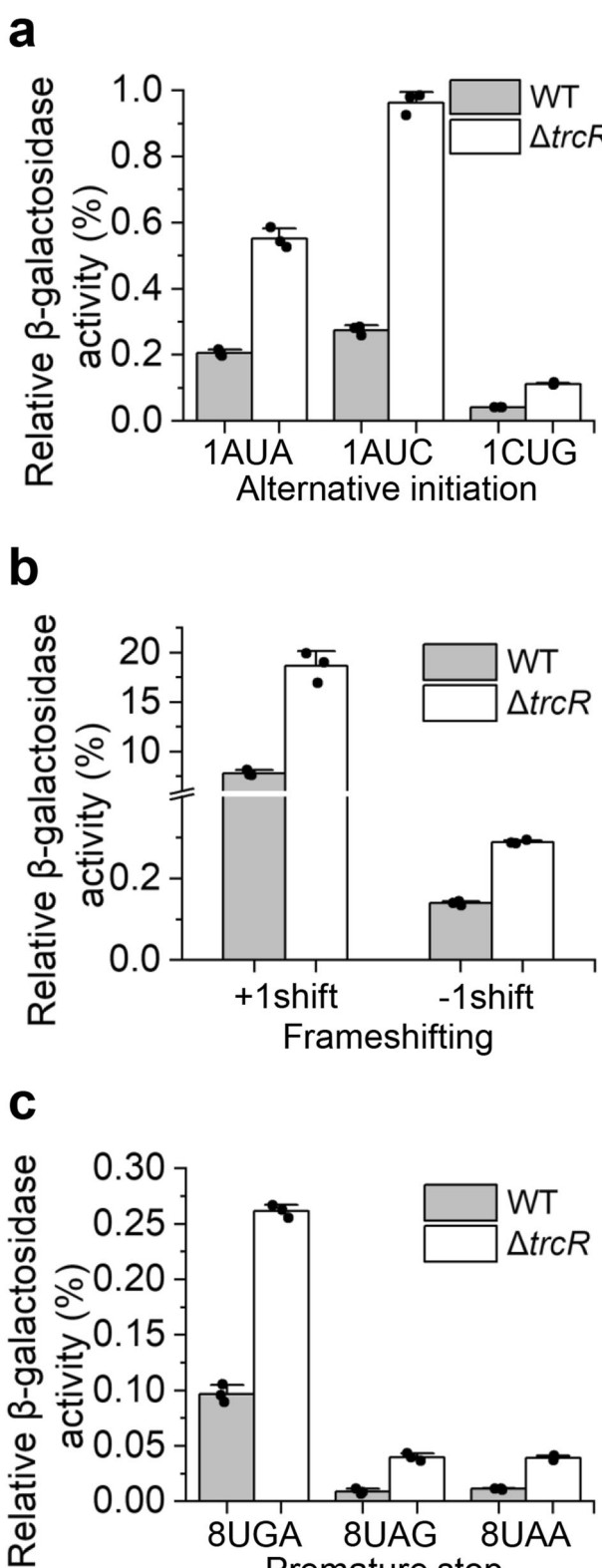

**Fig. 7 Effect of *trcR* deletion on translational fidelity.** Plasmids carrying *lacZ* and its derivatives with (**a**) alternative initiation codons (AUA, AUC and CUG), (**b**) frameshift mutations (+1 or -1) or (**c**) nonsense mutations (UGA, UAG and UAA) were transferred into the WT and Δ*trcR* mutant, respectively. The level of β-galactosidase activity was used to measure the fidelity of the translational machineries to translate mutant forms of *lacZ* mRNAs. Data were normalized to the β-galactosidase activity of the WT LacZ in either WT or Δ*trcR*. Data shown are the mean values ± S.D. (*n* = 3).

## Discussion

In the previous study, through BMAA-resistance mutants screening and genetic analysis, we found that tRNA t6A modification play an important role in translational control in *Anabaena*[19]. In the present study, by investigating other BMAA-resistance mutants, we identified a gene of unknown function and named here as *trcR*, with its product acting as a repressor to exert a global control on translational processes. TrcR belongs to the RHH family of transcription factors. Our DNA-binding assays, DNA footprinting, transcriptome, and transcriptional analyses using CFP as a reporter or qRT-PCR, all confirmed that TrcR is a transcription factor. The binding data obtained allowed us to deduce a consensus binding site for TrcR, which could be important for identification of TrcR regulated genes. Most of the TrcR binding sites identified in this study are located near, or overlap with, the promoter regions of the corresponding genes, consistent with its repressive role in gene regulation.

The deletion of *trcR* led to two major genetic consequences, the resistance to BMAA and the antibiotic Ksg, and derepression of the *trn* operon and several other genes involved in translation (Fig. 9). The reason for BMAA resistance of Δ*trcR* could be determined, which was caused by the derepression of *alr1538* encoding an efflux system responsible for the decrease of BMAA levels detected in the cells. However, the reason for Ksg resistance remains unknown. The inactivation of *alr1538* or any of those highly expressed genes identified in Δ*trcR* mutant could not alleviate the Ksg-resistance effect. Previously, we found that Ksg could cause a translational stress in *Anabaena*[19]. Thus, the Ksg resistance acquired by Δ*trcR* could be attributed to either an activated efflux pump or a modification of translational machineries such as ribosomes, making Ksg unable to bind efficiently to its targets. The lack of translational stress caused by Ksg in Δ*trcR*, in contrast to other antibiotics targeting translation, is consistent with such possibilities.

The identification of TrcR as the repressor of the *trn* operon and several other genes involved in translation represents an important advance in our understanding on the translational control in cyanobacteria (Fig. 9). TrcR is the missing link between translational stress induced by antibiotic treatment and the expression of the otherwise silenced *trn* operon in *Anabaena* as reported recently[8]. Ignacio Luque et al.[8] reported that the transcription of *trn* could be activated by antibiotics targeting the translational process, which increased cell competitiveness and survival in *Anabaena*. In this study, we found that the regulation of the *trn* operon was accomplished by TrcR. In addition to the *trn* operon present on a megaplasmid, several other chromosomal genes such as *all3526* (*rtcB*), *alr8077* (*rsgA*), *alr3303* (*rimK*) involved in translation are also repressed by TrcR. The expression of *rtcB* can be induced by the accumulation of damaged tRNAs upon translational stress in *S. typhimurium*[40], and RtcB plays a role in re-ligation of truncated 16 S rRNA upon stress relief in *E. coli*[30]. In *E. coli* and *S. typhimurium*, the expression of *rtcB* was regulated by the σ[41]-dependent transcriptional activator RtcR[40,42]. No homologous protein of RtcR in *Anabaena* could be identified, consistent with the absence of alternative sigma factors in cyanobacteria. Instead, our studies indicate that TrcR, belonging to a different family of transcription factors, has a function equivalent to that of RtcR.

Our results demonstrate that in *Anabaena*, the repression of TrcR on gene expression, including that of *trcR* itself, was relieved in the presence of antibiotics that inhibit the process of translational elongation (Cm, Sp and Sm). Although *trcR* transcriptional expression was derepressed under such conditions, the amount of TrcR protein decreased, which provided an explanation on the derepression of genes belonging to the TrcR regulon (Fig. 6b and c). A posttranslational regulation, such

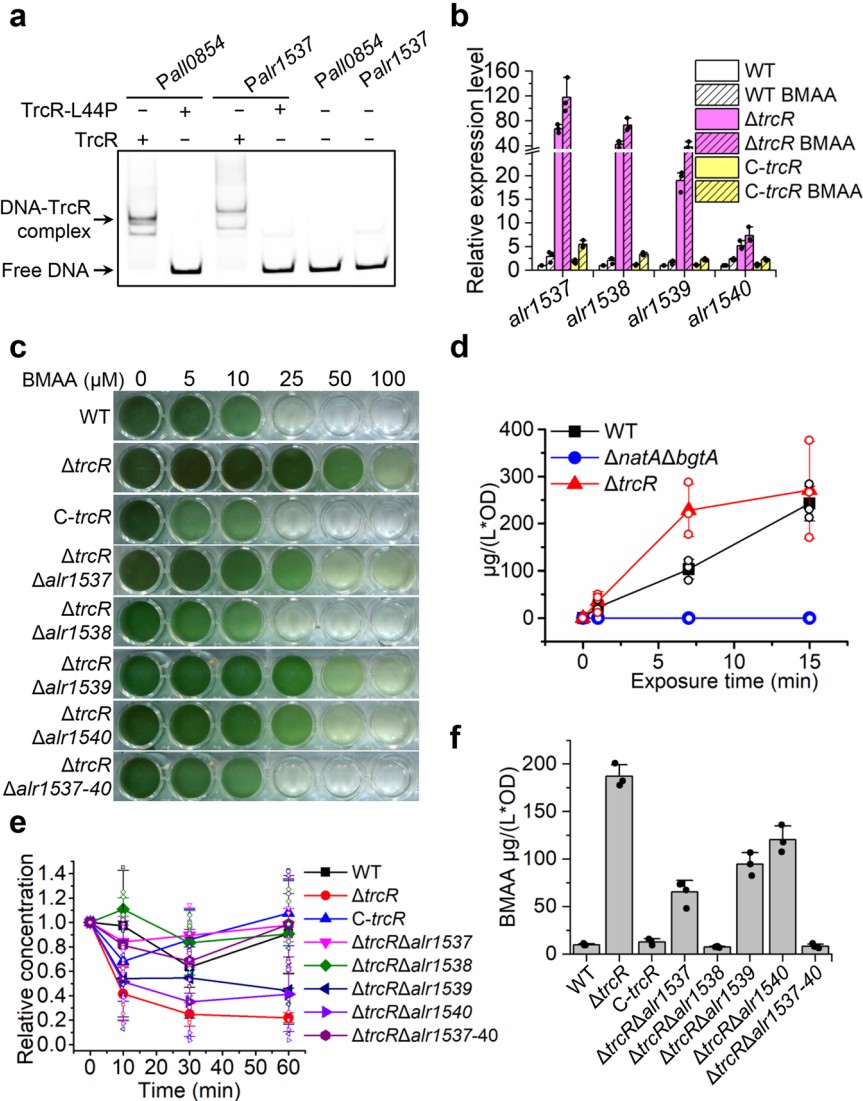

**Fig. 8 TrcR represses the expression of the operon *alr1537-alr1540* involved in BMAA export. a** EMSA testing the binding of TrcR and TrcR-L44P to the promoter region of *alr1537*. The promoter of *all0854* (*trcR*) was used as a control. **b** Transcription levels of *alr1537*, *alr1538*, *alr1539* and *alr1540* quantified by qRT-PCR in WT, Δ*trcR* and C-*trcR* with or without BMAA treatment. *n* = 3 biologically independent samples. The experiment was repeated three times for each sample. Data shown are the mean values ± S.D. **c** BMAA sensitivity of Δ*trcR*Δ*alr1537*, Δ*trcR*Δ*alr1538*, Δ*trcR*Δ*alr1539*, Δ*trcR*Δ*alr1540* and Δ*trcR*Δ*alr1537-40* tested in BG11 liquid medium containing different concentrations of BMAA. **d** Uptake of BMAA in WT, Δ*trcR* and Δ*natA*Δ*bgtA* at indicated time points. **e, f** Quantification of BMAA uptake or secretion in WT, Δ*trcR*, C-trcR, Δ*trcR*Δ*alr1537*, Δ*trcR*Δ*alr1538*, Δ*trcR*Δ*alr1539*, Δ*trcR*Δ*alr1540* and Δ*trcR*Δ*alr1537-40*. **e** Amount of intracellular BMAA quantified at indicated time points after transferring into the BMAA-free medium. **f** Amount of BMAA secreted into the supernatant after transferring into the BMAA-free medium for 60 min. Data shown in (**d**), (**e**) and (**f**) are mean values ± S.D from triplicates.

as proteolysis or translational inhibition of *trcR* mRNA, may occur to account for the decreased level of TrcR under translational stress. The signaling processes that regulates both the amount of TrcR and its DNA-binding activity remain to be understood.

In addition to its function in translational control, TrcR also regulates the expression of other genes, such as the *alr1537-alr1540* operon, among which *alr1538* encodes a DMT family protein at the core of BMAA export machinery. Therefore, TrcR has multiple targets involved in different functions (Fig. 9), consistent with our transcriptome data. TrcR, as well as its binding sites, are highly conserved in many unicellular and filamentous cyanobacterial species, but are missing in the group that forms branching filaments (Supplementary Fig. 16) and some marine picocyanobacteria such as *Prochlorococcus*. Two homologs of TrcR are found

in *Synechococcus* sp. PCC 7003 (WP_065713646.1) and *Synechococcus* sp. PCC 11901 (QCS49454.1), which are marine strains[43,44]. TrcR appears to represent a lineage of RHH regulators restricted to cyanobacteria, and two homologs found out of the cyanobacterial phylum correspond to those present in two cyanophages, annotated as *Nodularia* phages vB_NpeS-2AV2 and vB_NspS-kac65v151 in metagenomic data[45]. The transcriptional regulation of translation by TrcR represents an important mechanism for the coupling of the transcriptional and translational regulations in cyanobacteria, and its functional studies will open a new horizon for our understanding of the adaptation mechanisms in cyanobacteria. The resistance to certain antibiotics acquired by some of the mutants may also help us to understand the environmental effects of antibiotics increasingly present in water bodies as a consequence of human activities.

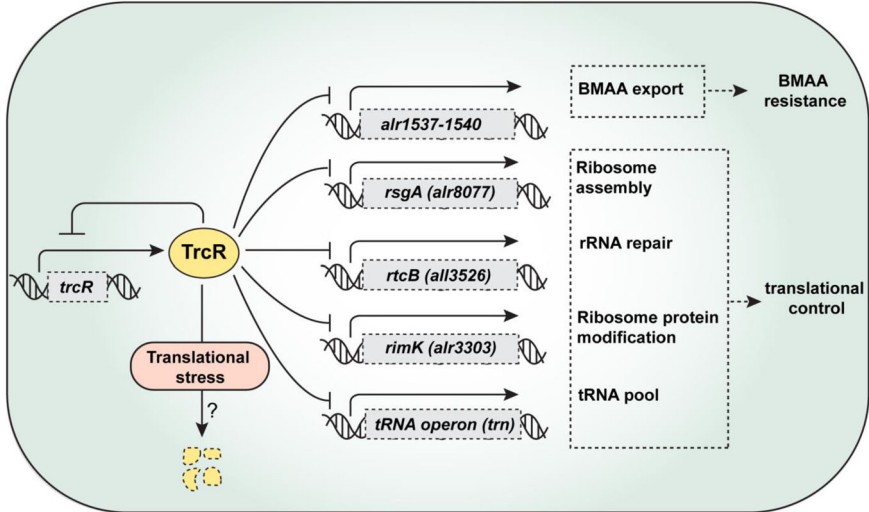

**Fig. 9 Cartoon illustrating the function of TrcR.** As an autoregulated transcriptional repressor, TrcR inhibits the transcription of several genes involved in translational control such as *rsgA*, *rtcB*, *rimK* and *trn* under normal laboratory culture conditions. However, the gene inhibition was relieved under translational stress induced by antibiotics, probably due to TrcR degradation through an unknown pathway. Besides the translational genes, TrcR also regulates the expression of an operon *alr1537-alr1540*, whose protein products are responsible for the export of BMAA.

## Methods

**Strains and growth conditions**. BG11 medium[46] was used to culture *Anabaena* and its derivatives using conditions as described[28]. All the cyanobacterial strains used in this study are described in Supplementary Table 2. For growth curve measurement, strains were cultivated in liquid BG11 medium with or without 25 μM BMAA, and the cell density was determined at $OD_{750}$. To test the sensitivity of *Anabaena* and its derivatives to BMAA, the tested strains were cultured (with a starting $OD_{750}$ of 0.075) in BG11 containing different concentrations of BMAA in the 24-well plates, followed by imaging from the bottom of plates after 7 days. The sensitivity of *Anabaen*a and its derivatives to antibiotics was tested as described in the previous study[19]. Briefly, the $OD_{750}$ of the cultures at exponential phase was adjusted to 1.0. After serial dilutions (1/2, 1/4, 1/8, 1/16 and 1/32), 3 μL of cell cultures were spotted onto BG11 agar plates containing specific antibiotics (chloramphenicol (Cm), kasugamycin (Ksg), streptomycin (Sm), spectinomycin (Sp) and penicillin G (PenG)) at different concentrations. The plates were imaged after 10 days of incubation. BMAA was purchased from Wuhan Chuanliu Bio-technology Co., Ltd. (Wuhan, Chian). Ksg was from Shanghai Yuanye Bio-technology Co., Ltd. (shanghai, China), and all the other antibiotics were purchased from Sigma-Aldrich Co., LLC. (St. Louis, MO, USA).

**Spontaneous BMAA-resistance mutants screening and genomic sequencing**. The methods for spontaneous BMAA-resistance mutants screening in the presence of 100 μM BMAA were described in our previous study[18]. Genomic DNA of WT or the mutants was extracted and sequenced by BGI (BGI company, Shenzhen, China) using the next-generation sequencing technique based on Illumina Hiseq 4000 system (Illumine, San Diego, CA, USA). The average sequencing depth was 9.8 million reads per sample with 97-99% coverage of the *Anabaena* genome. The details of the sequencing and data analysis methods were described previously[19].

**Electrophoretic mobility shift assays (EMSA)**. The strep-tagged TrcR and TrcR-L44P were heterologously expressed in *Escherichia coli* BL21 and purified for EMSA. DNA fragments of about 200 bp (predicted promoter regions) were labeled by fluorescent 6-carboxyfluorescein (FAM) tag at both ends. EMSA was carried out in 20 μL of EMSA buffer (1 mM DTT, 20% (v/v) glycerol, 0.1% (v/v) triton-100, 12 mM HEPES at pH 8.0, 4 mM Tris-Cl at pH 8.0, 60 mM KCl, 5 mM MgCl₂, 0.5 mM EDTA, 0.05 μg poly(dI:dC), 300 ng DNA and 0.2 μg TrcR or TrcR-L44P). After incubation at 30 °C for 30 min, samples were loaded on 5% (w/v) native polyacrylamide gels and ran in PAGE buffer (50 mM Tris-Cl at pH 8.0, 380 mM Glycine and 2 mM EDTA at pH 8.0) at 20 mA for 200 min. The gel was then imaged under excitation light of 490 nm that excited FAM. In this work, the promoter regions of studied genes were selected according to the RNA-Seq data published by Mitschke et al.[26].

**DNase I footprinting**. The method used for DNase I footprinting was adapted from that described by Wang et al.[47]. Here, purified TrcR and TrcR-L44P proteins were the same as that for EMSA, but the DNA fragments were labeled with FAM at only one end. The binding of TrcR to DNA fragments was performed in 40 μL reaction buffer (1 mM DTT, 20% (v/v) glycerol, 0.1% (v/v) triton-100, 12 mM HEPES at pH 8.0, 4 mM Tris-Cl at pH8.0, 60 mM KCl, 5 mM MgCl₂, 0.5 mM

EDTA, 500 ng DNA and 0.8 μg TrcR) at 30 °C for 30 min. The same reaction system without TrcR was used as a negative control. DNase I (Takara Biomedical Technology Co., Ltd, Beijing, China) was added into the reaction system to the final concentration of 0.005 U/μL. After 1 min of incubation at room temperature, 140 μL stop buffer (200 mM sodium acetate, 30 mM EDTA at pH 8.0 and 0.15% (w/v) SDS) was added into the reaction mixture. Then, the digested DNA was precipitated with ethanol and the dried pellet was dissolved in 10 μL TE buffer (10 mM Tris-HCl, 1 mM EDTA, pH 8.0). The samples were subsequently sequenced and analyzed by Tsingke Biotechnology Co., Ltd. (Beijing, China). The sequences protected by TrcR from DNase I digestion were further analyzed on the website MEME (https://meme-suite.org/meme/tools/meme).

**RNA preparation and transcriptome analysis**. WT and Δ*trcR* strains in triplicate were cultured to logarithmic phase in BG11. The cell samples were collected quickly by filtration and soaked in RNAlater (Wuhan Chuanliu Biotechnology Co., Ltd. Wuhan, China) for half an hour. Then, the RNA was isolated from these samples according to the hot phenol procedure[48]. After quality check by agarose gel electrophoresis, the RNA samples were sent to BGI for RNA-seq and data analysis. The details of the procedures for RNA-seq and data analysis were described in our previous publication[19].

**Quantitative real-time PCR (qRT-PCR)**. qRT-PCR was performed to compare the transcript levels of *all3526*, *alr3303*, *alr8077* and *trn* operon among WT, Δ*trcR* and C-*trcR* strains. All strains were cultured to exponential phase before being collected by rapid filtration. To detect changes in gene transcription levels upon Cm treatment, WT were cultured to exponential phase and a sub-lethal concentration (5 μg/mL) of Cm was added into the medium, then cell samples were collected at 0, 3, 6, 9, 24 and 48 h. For comparing the transcription levels of *alr1537-alr1540* with or without BMAA treatment, WT, Δ*trcR* and C-*trcR* were cultured to exponential phase. 25 μM of BMAA was added into the medium 24 h before sampling. Samples without BMAA served as control. All strains mentioned above were cultured in triplicate and total RNA of collected samples was extracted for further qRT-PCR analysis.

The kit HiScript II QRT SuperMix for qRCR ( + gDNA wiper) (Vazyme Biotech Co., Ltd, Nanjing, China) was used for reverse transcription following the manufacturer's instruction. qRT-PCR was performed by C1000 Touch Thermal Cycler (Bio-Rad Laboratories, Inc., Hercules, CA, USA) using ChamQ SYBR qPCR Master Mix (Vazyme Biotech Co., Ltd) with three technical replicates for each sample. The transcript level of *allrs04* encoding RNase P served as the internal control. All primers used for qRT-PCR were listed in Supplementary Table 3. The relative transcription levels of genes were obtained according to the $2^{-\Delta\Delta CT}$ calculation method[49]. Data were normalized by the transcription level of the corresponding genes in the WT without BMAA treatment.

**Western blot**. To evaluate the expression of TrcR in *Anabaena* under the stress caused by the presence of Cm, Ksg or PenG, WT cells in triplicate were cultured to exponential phase, and then antibiotics of sub-lethal concentration (Cm 5 μg/mL, Ksg 5 μg/mL or PenG 1 μg/mL) were added into the medium, respectively. After incubation for 0, 3, 6, 9, 24 and 48 h, cells from 30 mL cultures were collected by

filtration for western blot analysis following the procedure as described[50]. The antibody was prepared by injecting recombinant TrcR protein into rabbit (Mabnus Biotech Co., Ltd, Wuhan. China).

**Translational fidelity test and β-galactosidase activity measurement**. Translational fidelity test was performed as previously described[19]. Briefly, 9 plasmids (pTac-lacZ, pTac-lacZ1ATA, pTac-lacZ1ATC, pTac-lacZ1CTG, pTac-lacZ8TAA, pTac-lacZ8TAG, pTac-lacZ8TGA, pTac-lacZ+1shift and pTac-lacZ-1shift) that carried a series of *lacZ* derivatives as reporters were constructed. pTac-lacZ carried the wild-type *lacZ* gene. In pTac-lacZ1ATA, pTac-lacZ1ATC and pTac-lacZ1CTG, the initiation codon AUG of *lacZ* was changed into AUA, AUC and CUG, respectively. In pTac-lacZ+1shift and pTac-lacZ-1shift, frameshifting mutation ( + 1 or -1) was created after the seventh codon. For pTac-lacZ8TAA, pTac-lacZ8TAG and pTac-lacZ8TGA, the eighth codon of *lacZ* was replaced by stop codon UAA, UAG and UGA, respectively. The transcription of *lacZ* and its derivatives was under the control of the *tac* promoter[51]. All plasmids were transformed respectively into *Anabaena* or its derivatives. The translational fidelity of these strains was characterized by β-galactosidase (LacZ) activity[19].

**Measurement of BMAA uptake and export**. To test the ability of BMAA uptake, WT, Δ*trcR* and Δ*natA*Δ*bgtA* were cultured to the exponential phase and exogenous BMAA was added into the medium at a final concentration of 50 μM. 20 mL of cell cultures were collected at time 0 (right before BMAA addition), 1, 7 and 15 min (after BMAA addition), respectively. Cell samples were harvested by filtration and washed by BMAA-free medium to remove residual BMAA as soon as possible. The samples were stored at -80 °C for further processing.

To prepare the samples for detecting BMAA export, *Anabaena* or its derivatives was first incubated with 50 μM BMAA for 15 min, followed by washing with BMAA-free medium for three times. Cell samples from 20 mL cell cultures were harvested by filtration after 0, 10, 30 and 60 min and washed by BMAA-free medium. After 60 min, 500 μL medium of which the cells were removed by 0.2 μm filter was collected to determine the concentration of BMAA exported from the cells. All samples were stored at −80 °C for further processing.

The technique of ultra-high performance liquid chromatography (UPLC) with tandem mass spectrometry detection (UPLC-MS/MS), combined with derivatization using 6-aminoquinolyl-N-hydroxysuccinimidyl carbamate (AQC), was used for BMAA detection[52]. Sample preparation, derivatization and the UPLC-MS/MS condition were described in detail in our previous publication[18].

**Construction of plasmids and mutants**. All the markerless deletion mutants used in this study were constructed using the Cpf1 genome editing system[53,54]. The plasmids pCpf1b-Mall0854R246, pCpf1b-Malr1537R372, pCpf1b-Malr1538F404, pCpf1b-Malr1539R169, pCpf1b-Malr1540R179 and pCpf1b-Malr1537-40R674 were used to construct Δ*trcR*, Δ*alr1537*, Δ*alr1538*, Δ*alr1539*, Δ*alr1540* and Δ*alr1537-40* respectively. The method for the construction of these plasmids followed the previously described procedures[54]. Briefly, the plasmid pCpf1b-sp[54] was linearized by restriction enzyme *Bam*H I and *Bgl* II. The upstream and downstream fragments used for homologous recombination were amplified by PCR from genomic DNA of *Anabaena* PCC 7120. The linearized pCpf1b-sp, the upstream and downstream homologous fragments, were all ligated by the ClonExpress MultiS One Step Cloning Kit (from Vazyme Biotech Co.,Ltd; Nanjing, China) to form a precursor plasmid. Finally, the precursor plasmid was digested by *Aar* I, and the corresponding guide sequence was ligated into the digested precursor plasmid to complete the construction. The primers used for plasmid construction were listed in Supplementary Table 4.

To construct the mutants used in this study, the plasmids mentioned above were transformed into *Anabaena* PCC 7120 or Δ*trcR* by conjugation[41,55]. Positive colonies were screened by using 5 μg/mL spectinomycin and 2.5 μg/mL streptomycin. To get the markerless mutants, positive colonies were continuously screened by using 5% sucrose in BG11 agar plates to remove the Cpf1-based plasmids follow the published procedure[54]. The genotypes of all the constructed mutants were verified by PCR.

The vector for constructing the transcriptional fusion plasmids p$_{trcR}$CFP, p$_{alr1537}$CFP, p$_{alr3301}$CFP, p$_{alr8077}$CFP, p$_{all3526}$CFP and p$_{trn}$CFP was amplified from the plasmid pRLRBS-mTur by primers PpCT-R2979 and PV_16. pRLRBS-mTur, modified from the shuttle plasmid pRL25T[56], carries the ORF of *cfp*. The promoter regions of these genes of interests were defined according to the published RNA-Seq data[26]. The promoter regions amplified from *Anabaena* PCC 7120 genomic DNA were inserted, respectively, into the pRLRBS-mTur vector through ClonExpress II One Step Cloning Kit (Vazyme Biotech Co.,Ltd; Nanjing, China) to complete the construction.

To construct the complementation plasmid pRL-Call0854, the vector was amplified by PCR from the replicative plasmid pCT[27] with primers PpCT-R2979 and PpCT-F3530. The gene *all0854* with its native promoter region was amplified by PCR from the genomic DNA of *Anabaena* PCC 7120. Finally, the vector and the gene fragment were ligated via ClonExpress II One Step Cloning Kit to complete the construction.

The transcriptional fusion plasmids or the complementation plasmid were transformed into *Anabaena* PCC 7120 or its derivatives by conjugation to get the corresponding transcriptional fusion strain or the complemented strain[41,55].

**Statistics and Reproducibility**. In qRT-PCR assays, the relative transcriptional levels of the genes of interests were calculated through $2^{-\Delta\Delta CT}$ calculation method based on triplicate data[49]. The growth curves data were collected from two parallel cultures and presented as mean values. The relative fluorescence intensity of CFP reporter from stains with transcriptional fusions and the relative integrated density of the Western blot bands were quantified by ImageJ v1.51j8. For BMAA uptake and export detection assays, all the data were collected from 3-6 biological replicates. In this study, all the statistics analysis was carried out by SPSS v20.0 or Origin v2022, and the data are presented as mean±S.D. (standard deviation).

**Reporting summary**. Further information on research design is available in the Nature Portfolio Reporting Summary linked to this article.

## Data availability

RNA-seq data have been deposited at the Gene Expression Omnibus (GEO) database under the accession number GSE218875. The source data for Figs. 1b, 4d, 5a, b, 6a, b, 7, 8b, d–f were shown in Supplementary Data 4–13 respectively. The blot/gel images in Fig. 2c–e, Fig. 4a–c, Fig. 6b, Fig. 8a and Supplementary Figs. 3a, b, d, 5b, 14b were edited from the original pictures shown in Supplementary Figs. 17-23 respectively. Plasmid pCpf1b-sp was deposited at Addgene with ID number #122188. Plasmid pCT was submitted to GenBank with ID number MK948095. All other data are available from the corresponding author upon request.

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

## Acknowledgements

This work was supported by Youth Program of National Natural Science Foundation of China (grant No. 31900028), The Featured Institute Service Projects from the Institute of Hydrobiology, the Chinese Academy of Sciences [Y85Z061601] and the State Key Laboratory of Freshwater Ecology and Biotechnology [2022FBZ04]. Funding for open access charge: The Featured Institute Service Projects from the Institute of Hydrobiology, the Chinese Academy of Sciences. We thank the staff from the Analysis and Testing Center of the Institute of Hydrobiology, Chinese Academy of Sciences for help with qRT-PCR analysis and BMAA measurement using UPLC-MS/MS.

## Author contributions

C.-C.Z., J.-Y.Z. and Z.-Q.W. planned and designed the research; Z.-Q.W. performed the experiments; Z.-Q.W., Y.Y., J.-Y.Z., X.Z. and C.-C.Z. analyzed the data; Z.-Q.W., Y.Y. and C.-C.Z. wrote the manuscript. Y.Y. and C.-C.Z. edited the manuscript.

## Competing interests

The authors declare no competing interests.
