## [Peer Review File · Communications Biology]

Reviewers' comments:

Reviewer #1 (Remarks to the Author):

The manuscript by Wang et al delves into the role of a transcriptional regulator TrcR in regulating translation as well in Anabaena PCC 7120. To substantiate their claims several experiments have also been conducted, which has helped answer some of the questions but has raised several others. Possibly due to the global regulatory role of TrcR, all the observed phenomena cannot be explained. In Anabaena 7120, a few global transcriptional regulators such as NtcA-PipX and LexA are already known, and this work does initiate further thinking in terms of translational global regulator, however,. The major concerns/queries are as follows:

1. Lines 104-107: If mutation is same for M19 and M20, then their Resistance to BMMA should also have been similar. However, M18 n M20 are showing similar resistance, while M19 is highly sensitive. How can this be explained?
2. Line 108-112 and Fig. 1: There is no mention of the protein levels of TrcR in the complementation studies i.e. whether they are expressed to similar levels as in WT strain. Secondly, differential sensitivity is observed for WT, M20-CtrcR and C-trcR, with C-trcR being most sensitive and M20-CtrcR least. How do you explain the differences in sensitivity?
3. In Fig. 2B It would have been more appropriate to test the binding with another promoter fragment which lacks the conserved DNA motif instead of a region corresponding to the gene or downstream of the gene
4. In Fig. 2D and E Could DNA3 and DNA4 compete with each other, since they overlap. That should be included
5. Line 152: The experiments were done only with M20, What about M18? Were these also tested and if so what new results did they throw up?
6. Lines 244-246: What was the basis for this presumption that the gene overexpression could be the cause for Ksg resistance? Based on the transcriptomics data, can no speculation be done. Schuwirth et al 2006 have given some insights into probable causes of Ksg resistance. Can some clue be taken from their paper. Something on similar lines is mentioned in discussion, however, it needs to be elaborated.
7. The observation that trcR transcript levels increase but protein levels decrease in the presence of Cm (para starting at line 251) raises a few questions: (i) Have you checked if it is the stability of the protein which is affected or synthesis itself has gone down, which can be done using 35S labelled methionine. (ii) Secondly, you have only looked at increased gene expression when TcrR levels decrease, so does it result in decreased protein level of those genes or they also behave like tcrR showing increased transcript but lower protein levels. This needs to be ascertained.
8. Lines 262-263 You need to differentiate between translational and post translational control
9. Lines 276-278: If the deletion of tcrR leads to decreased translational fidelity as stated, then it should be lethal to cells as it may end up synthesising incorrect proteins, some of which may be essential. And this function may not be related to its transcriptional repressor role
10. Fig. 8B: Any reason why the transcript levels are lowest for the last gene of the operon in delta trcR mutant, while in WT cells, they look almost same? How does it translate at protein level?
11. Any proof that deletion of a gene in the operon (alr1537-1540) did not affect the transcription of the subsequent genes of the operon
12. Lines 384-385: This observation is too farfetched and should be toned down
13. General comment: Since all signifies left orientation and alr right orientation, this must be maintained in schematic representation. It is OK to use the complementary sequence for A, but fr B the direction should be maintained. Else it should be mentioned that for ease of understanding the orientation is changed

Reviewer #2 (Remarks to the Author):

Wang et al. identify All0854, here renamed to TrcR, as a transcription factor in the cyanobacterium *Anabaena* PCC 7120 that represses its own gene and several other genes involved in the response to translational stress.

Among these genes controlled by TrcR is a long cluster of more than 20 tRNA genes, also called the L-array. All0854 was not previously characterized and the function and regulation of the L-array had been truly enigmatic.

The study provides substantial conceptual advance, is carefully conducted and should be published.

I have the following comments:

Results.

L131-132: "a region of 25 bp from -50 to -25 (relative to the translational start site of *trcR*) is essential for TrcR binding" Because this is about transcriptional regulation, here information should be added where this motif is located with regard to the actual transcription start site.

The previous genome-wide mapping (Table S1 in PMID: 22135468) established it to be located on pos. 983055r, the penultimate nucleotide in the here identified motif, the "a", in 5'-ATACTACACTTGTATTaC-3', from which "TACACT" was inferred as the respective -10 element. In other words, the binding site overlaps directly the start site of transcription.

Therefore, the location is very consistent with the later made conclusion of TrcR binding leading to autorepression, because it blocks access to the most important elements of its promoter.

Discussion.

The input signal for TrcR is the most important question that remains. I understand that it's identification is beyond the scope of the current study, but it is may be possible to briefly discuss what other members of this group of transcription factors are sensing and if that is possibly relevant.

TrcR appears to be highly conserved in many different cyanobacteria. Please add an explicit statement whether TrcR represents a lineage of RHH regulators that is exclusive to cyanobacteria, or not.

Looking at it, I found outside of the phylum closer homologs only in two cyanophages, *Nodularia* phages vB_NpeS-2AV2 and vB_NspS-kac65v151 and in several metagenomes where the taxonomic assignments may be incorrect. I suggest to include a more complete phylogenetic tree to answer this question. Given the function of TrcR in a process as fundamental as translation and the avoidance of inhibition by antibiotics, I find it stunning if it were restricted to cyanobacteria.

L384: There are homologs in *Synechococcus* sp. PCC 7003 (WP_065713646.1) and *Synechococcus* sp. PCC 11901 (QCS49454.1), which are marine strains (Commun Biol 3, 215 (2020), J Mol Evol (1999) 48:723-739).

This should be mentioned for the sake of correctness. Consequently, homologs are absent from the vast majority of marine cyanobacteria, but not all.

Methods.

Please mention briefly or cite an appropriate reference for the transformation procedure.

Although the language use and style is generally really good, I spotted a few instances that need correction:

L48/49: "One of them..." improve wording by saying "One of these types"

L82: "BMAA-resistant" should be "BMAA-resistance"

L115: "databanks" should be two words

L168: The plural form would be more appropriate instead of "EMSA was performed.."; these were many experiments.

L435-436: "but the DNA fragments was labelled" requires the plural form

Table 1: Insert in the table legend "be" between "may" and "transcribed".

Reviewer #1 (Remarks to the Author):

The manuscript by Wang et al delves into the role of a transcriptional regulator TrcR in regulating translation as well in Anabaena PCC 7120. To substantiate their claims several experiments have also been conducted, which has helped answer some of the questions but has raised several others. Possibly due to the global regulatory role of TrcR, all the observed phenomena cannot be explained. In Anabaena 7120, a few global transcriptional regulators such as NtcA-PipX and LexA are already known, and this work does initiate further thinking in terms of translational global regulator, however, The major concerns/queries are as follows:

1. Lines 104-107: If mutation is same for M19 and M20, then their Resistance to BMMA should also have been similar. However, M18 n M20 are showing similar resistance, while M19 is highly sensitive. How can this be explained?

According to our experiences, several mutation sites are frequently found over the chromosome, when mutants were isolated, even when we used the WT strain just before the screening as the control for sequence comparison. This is true for our BMAA resistant mutants reported here are elsewhere, and also true for other mutants we did in other projects (see, for example, Xing et al., PNAS 119:e2207963119, 2022, in addition to those in patU3, some mutations could occur elsewhere following suppressor screening, and different alleles show differences in their phenotypes). That is why we need to reconstruct the mutant, followed by complementation to ascertain that the gene is responsible for the phenotype. The screening process is used for initial screen, and all late step studies rely on reconstructed mutants, for phenotypic analysis.

In the current case, we only did whole genome sequencing for M20, and others were checked by PCR and sequencing, to minimize the workload. Because multiple genes are involved in BMAA resistance, additional mutations may occur.

2. Line 108-112 and Fig. 1: There is no mention of the protein levels of TrcR in the complementation studies i.e. whether they are expressed to similar levels as in WT strain. Secondly, differential sensitivity is observed for WT, M20-CtrcR and C-trcR, with C-trcR

being most sensitive and M20-CtrcR least. How do you explain the differences in sensitivity?

We thank the reviewer for this question, which is dosage-dependent effect as we explain below. Indeed, as shown below by western blot, the complemented strain has a higher expression of TrcR than the WT (the WT allele is on a replicative plasmid), consistent with the higher sensitivity shown in fig. 1. Although we did not perform western for M20-CtrcR, the sensitivity of it to BMAA is recovered back to the WT level. Either the expression level is as in the WT, or it is higher, but the presence of both the WT and the mutant version may interfere partly its function since the transcription factor forms dimers. What is important is that the presence of the WT copy carried on a plasmid can rescue the phenotype.

3. In Fig. 2B It would have been more appropriate to test the binding with another promoter fragment which lacks the conserved DNA motif instead of a region corresponding to the gene or downstream of the gene.

In fact, such data is already present in our manuscript. We tested several other genes' promoter fragments and got negative binding results, for example the promoter fragment of *all5040* that lacks the conserved DNA motif. The result is shown in Figure 4A and lines 181-183.

4. In Fig. 2D and E Could DNA3 and DNA4 compete with each other, since they overlap. That should be included.

That is a good suggestion. And we added this results in supplementary Figure S3A and B. And added the sentence 'The competition of unlabeled DNA4 with FAM-labeled DNA3, or of unlabeled DNA3 with FAM-labeled DNA4, also occurred; .' in lines 131-134.

5. Line 152: The experiments were done only with M20, What about M18? Were these also tested and if so what new results did they throw up?

We did not do this experiment in M18. The use of $\Delta trcR$, reconstructed with a clean genetic

background ought to be enough to prove that TrcR is an autorepressor. In WT the transcriptional fusion *cfp* cannot be expressed but in $\Delta trcR$ it can. The expression of the fused *cfp* in M20 is just additional evidence to prove that.

6. Lines 244-246: What was the basis for this presumption that the gene overexpression could be the cause for Ksg resistance? Based on the transcriptomics data, can no speculation be done. Schuwirth et al 2006 have given some insights into probable causes of Ksg resistance. Can some clue be taken from their paper. Something on similar lines is mentioned in discussion, however, it needs to be elaborated.

Since TrcR is a repressor, it was one of the possibilities to test. Our results ruled out this possibility, and the present data cannot help us to provide an adequate explanation. In the paper of Schuwirth *et al*, the authors analyzed the Ksg binding site in ribosome by X-ray structure. And their results demonstrate that inhibition by Ksg and Ksg resistance are closely linked to the structure of the mRNA at the junction of the peptidyl-tRNA and exit-tRNA sites in the ribosome. That indicates that Ksg resistance is related to ribosome structure. In this study we found some TrcR regulated genes such as *alr8077 (rsgA)* is related to ribosome assembly and it was reported that the deletion of *rsgA* in *E.coli* would affect the structure of ribosomes. Therefore, we tested if Ksg resistance of $\Delta trcR$ is related to the genes repressed by *trcR*. However, the double mutants (lines 252-258 and Figure S11) and the deletion of these genes in $\Delta trcR$ has no influence on Ksg resistance. Therefore, we can only discuss about alternative possibilities: maybe like BMAA, Ksg could be exported efficiently in $\Delta trcR$, the ribosome structure changed by some genes regulated directly by TrcR, or it's part of a global impact of the genes under control by TrcR, and maybe Ksg is metabolized or modified in $\Delta trcR$. We need further investigations to make it clear. Su a global regulator, we cannot reveal all of its functions in the current studies.

7. The observation that *trcR* transcript levels increase but protein levels decrease in the presence of Cm (para starting at line 251) raises a few questions: (i) Have you checked if it is the stability of the protein which is affected or synthesis itself has gone down, which can be done using 35S labelled methionine. (ii) Secondly, you have only looked at increased gene expression when TrcR levels decrease, so does it result in decreased protein level of those genes or they also behave like *trcR* showing increased transcript but

lower protein levels. This needs to be ascertained.

Yes, the reviewer is right! There are two reasons for the decrease of TrcR protein level. One is that the stability of it is affected under that Cm treatment condition, and it may be degraded by some protease. The other one is that the translation of *trcR* mRNA is inhibited. We are continuing to work out the underlying mechanism, but as we say above, we cannot pretend to understand everything in a single publication. For the second suggestion, although we did not check the protein levels of genes controlled by TrcR, However, based on the results described in the section 'TrcR regulates the expression of *alr1537-alr1540* responsible for BMAA export and resistance', these protein levels must be upregulated. The export of BMAA needs functional exporter. When the transcriptional level of this operon increases, the export rate of BMAA is raised. So, the protein levels of these genes, at least for the *alr1537-alr1540* operon, must increase, which is different from TrcR that has an autoregulation function.

8. Lines 262-263 You need to differentiate between translational and post translational control

I agree with this comment of the reviewer. There are two possible reasons for the decrease of TrcR protein level. One is synthesis itself has gone down, the other is that TrcR is unstable in the condition of Cm treatment. The former belongs to the category of translational control, however the latter belongs to post translational control. We changed the sentence to 'These results also suggest the existence of a translational or a posttranslational regulation of TrcR for the control of its protein level.' in lines 270-271.

9. Lines 276-278: If the deletion of *trcR* leads to decreased translational fidelity as stated, then it should be lethal to cells as it may end up synthesising incorrect proteins, some of which may be essential. And this function may not be related to its transcriptional repressor role

Many studies in both bacteria and eukaryotes show that a certain level of translation fidelity deficiency does not result in cell death, for example the study of Chalancon *et al.*, 2016. And in our previous work, we found that the defect of tRNA t⁶A modification could also result in the deficiency of translational fidelity in *Anabaena*, however the deletion of genes responsible for tRNA t⁶A modification can only affect the growth of *Anabaena*, without

leading to cell death. Many of the regulation systems are required for adaptation or fitness, and translational fidelity too. For important processes, such as translation, multiple control system exists for regulatory functions, not necessarily for cell survival.

References

- Chalancon, G., *et al.*, 2016. Global translational impacts of the loss of the tRNA modification t(6)A in yeast. *Microb. Cell.*, 3, 29–45.
- Wang and Zhang, 2022. A tRNA t⁶A modification system contributes to the sensitivity towards the toxin β -N-methylamino-L-alanine (BMAA) in the cyanobacterium *Anabaena* sp. PCC 7120. *Aquatic Toxicol.*, 245:106121.

10. Fig. 8B: Any reason why the transcript levels are lowest for the last gene of the operon in delta *trcR* mutant, while in WT cells, they look almost same? How does it translate at protein level?

For the first question, there are two possible theories to analyze why the mRNA levels of genes within one operon are different. In bacteria, it is very common to find that some consecutive genes within polycistronic operons show a decay behavior (Marc Güell *et al.*, 2009), which is consistent with genetic polarity. The other possible reason is that after transcription of the whole operon, the primary mRNA transcribed as an operon was processed into segments and the segments was with different stability (Chenggang Xu *et al.*, 2015). Genetic regulation mechanisms are multiple and complex, subsequent study may hopefully reveal more, but the current manuscript is certainly not the end of this scientific journey.

Reference

- Guell, M., *et al.*, 2009. Transcriptome complexity in a genome-reduced bacterium. *Science* 326, 1268 – 1271.
- Xu, C.G., *et al.*, 2015. Cellulosome stoichiometry in *Clostridium cellulolyticum* is regulated by selective RNA processing and stabilization. *Nat. Commun.*, 6, 6900.

11. Any proof that deletion of a gene in the operon (*alr1537-1540*) did not affect the transcription of the subsequent genes of the operon

We made a markerless deletion of the ORF by CRISPR-Cpf1, without leaving any

resistance marker. Such a deletion should have no influence on the transcription of subsequent genes of the operon.

12. Lines 384-385: This observation is too farfetched and should be toned down

In the revised version, we replaced 'therefore, TrcR is a global transcriptional regulator (Figure 9), consistent with our transcriptome data.', by 'Therefore, TrcR has multiple targets involved in different functions (Figure 9), consistent with our transcriptome data'. See lines 388-389.

13. General comment: 6 Since all signifies left orientation and alr right orientation, this must be maintained in schematic representation. It is OK to use the complementary sequence for A, but fr B the direction should be maintained. Else it should be mentioned that for ease of understanding the orientation is changed

Thanks for the good comment. In order to make readers better understand, we show the 5' and 3' ends and the transcriptional direction following the TSS of *trcR* in the Schematic Illustration of Figure 2B.

Reviewer #2 (Remarks to the Author):

Wang et al. identify All0854, here renamed to TrcR, as a transcription factor in the cyanobacterium *Anabaena* PCC 7120 that represses its own gene and several other genes involved in the response to translational stress.

Among these genes controlled by TrcR is a long cluster of more than 20 tRNA genes, also called the L-array. All0854 was not previously characterized and the function and regulation of the L-array had been truly enigmatic.

The study provides substantial conceptual advance, is carefully conducted and should be published.

I have the following comments:

Results.

L131-132: "a region of 25 bp from -50 to -25 (relative to the translational start site of *trcR*) is essential for TrcR binding" Because this is about transcriptional regulation, here

information should be added where this motif is located with regard to the actual transcription start site.

The previous genome-wide mapping (Table S1 in PMID: 22135468) established it to be located on pos. 983055r, the penultimate nucleotide in the here identified motif, the "a", in 5'-ATACTACACTTGTATTaC-3', from which "TACACT" was inferred as the respective -10 element. In other words, the binding site overlaps directly the start site of transcription.

Therefore, the location is very consistent with the later made conclusion of TrcR binding leading to autorepression, because it blocks access to the most important elements of its promoter.

Thanks to the reviewer, and that is a very good suggestion. In the revised version, we displayed the -10 box region and the TSS in Figure 2B, and added this sentence 'Based on the RNA-Seq results⁴⁵, the -10 box region 'TACACT' and transcription start site (TSS) of *trcR* overlaps with TrcR binding site, (Figure 2B), suggesting that TrcR could have an autorepression function. To confirm this hypothesis' in lines 143-145.

Discussion.

The input signal for TrcR is the most important question that remains. I understand that it's identification is beyond the scope of the current study, but it is may be possible to briefly discuss what other members of this group of transcription factors are sensing and if that is possibly relevant.

Thanks. We also tried to find the input signal for TrcR. The functions of the proteins belong to this RHH family are very different. They are involved in the uptake of metals, amino-acid biosynthesis, cell division, the control of plasmid copy number, the lytic cycle of bacteriophages and, perhaps, many other cellular processes. The RHH motif can be present within the amino-acid sequence of a protein, either as an isolated RHH motif or as part of larger proteins that have additional domains located on either side of the RHH DNA-binding domain (Eric R. Schreiter and Catherine L. Drennan, 2007). Some RHH family proteins are responded for effectors but the effectors are very different between different proteins, including metal ions, proteins and some small molecules. We analyzed the amino acids sequence of TrcR, RHH motif is at the N terminal. We carried out BLAST using its C terminal sequence, but we can not find any clue. So, it is very hard to even speculate

at this stage.

Reference

Eric R. Schreiter and Catherine L. Drennan, 2007. Ribbon–helix–helix transcription factors: variations on a theme. *Nat. Rev. Microbiol.*, **5**, 710–720.

TrcR appears to be highly conserved in many different cyanobacteria. Please add an explicit statement whether TrcR represents a lineage of RHH regulators that is exclusive to cyanobacteria, or not. Looking at it, I found outside of the phylum closer homologs only in two cyanophages, *Nodularia* phages vB_NpeS-2AV2 and vB_NspS-kac65v151 and in several metagenomes where the taxonomic assignments may be incorrect. I suggest to include a more complete phylogenetic tree to answer this question. Given the function of TrcR in a process as fundamental as translation and the avoidance of inhibition by antibiotics, I find it stunning if it were restricted to cyanobacteria.

We would like to thank the reviewer for these insightful comments. Following this suggestion, we made a more complete phylogenetic analysis (see the new tree in Supp Fig. S16). The corresponding text was also amended as follows in the revised version: 'TrcR, as well as its binding sites, are highly conserved in many unicellular and filamentous cyanobacterial species, but are missing in the group that forms branching filaments (Figure S16) and some marine picocyanobacteria such as *Prochlorococcus*. Two homologs of TrcR are found in *Synechococcus* sp. PCC 7003 (WP_065713646.1) and *Synechococcus* sp. PCC 11901 (QCS49454.1), which are marine strains^{40,41} (Commun Biol 3, 215 (2020), J Mol Evol (1999) 48:723–739). TrcR appears to represent a lineage of RHH regulators restricted to cyanobacteria, and two homologs found out of the cyanobacterial phylum correspond to those present in two cyanophages, annotated as *Nodularia* phages vB_NpeS-2AV2 and vB_NspS-kac65v151 in metagenomic data⁴². Lines 389-397

For comparison, this is the previous text before revision: 'TrcR, as well as its binding sites, are highly conserved in many unicellular and filamentous cyanobacterial species, but are missing in the group that forms branching filaments (Figure S16) and marine (such as *Prochlorococcus* and *Synechococcus*). These observations suggest that TrcR may have been evolved in order for cyanobacteria to cope with more complex freshwater environments'.

L384: There are homologs in *Synechococcus* sp. PCC 7003 (WP_065713646.1) and *Synechococcus* sp. PCC 11901 (QCS49454.1), which are marine strains (Commun Biol 3, 215 (2020), J Mol Evol (1999) 48:723–739).

This should be mentioned for the sake of correctness. Consequently, homologs are absent from the vast majority of marine cyanobacteria, but not all.

See the answer for the above comment. We added this part in the revised text.

Methods.

Please mention briefly or cite an appropriate reference for the transformation procedure.

We added two cited references in the section 'Construction of plasmids and mutants' of supplementary data in lines 17 and 36.

Elhai, J., Vepriksiy, A., Muro-Pastor, A. M., Flores, E. & Wolk, C. P. Reduction of conjugal transfer efficiency by three restriction activities of *Anabaena* sp. strain PCC 7120. *J Bacteriol* **179**, 1998-2005 (1997).

Elhai, J. & Wolk, C. P. Conjugal transfer of DNA to cyanobacteria. *Methods Enzymol* **167**, 747-754 (1988).

Although the language use and style is generally really good, I spotted a few instances that need correction:

L48/49: "One of them..." improve wording by saying "One of these types"

L82: "BMAA-resistant" should be "BMAA-resistance"

L115: "databanks" should be two words

L168: The plural form would be more appropriate instead of "EMSA was performed.."; these were many experiments.

L435-436: "but the DNA fragments was labelled" requires the plural form

Table 1: Insert in the table legend "be" between "may" and "transcribed".

In the new version lines 52-53: "One of them..." was changed to "one of these types"

In the text all 'BMAA-resistant' was changed to 'BMAA-resistance'.

In the new version line 119: 'databanks' was changed to 'data banks'.

REVIEWERS' COMMENTS:

Reviewer #1 (Remarks to the Author):

The author has clarified the queries raised and also introduced certain sentences for better clarity. The manuscript reads well and gives a good information on the repressor. The work should be published

Reviewer #2 (Remarks to the Author):

This work provides substantial conceptual advance and has been carefully conducted. During the revision, all my previously raised points have been addressed very well and the paper should be published.

I just have the following two minor points that still need correction:

L392: Please correct spelling of "Prochorococcus" to "Prochlorococcus".

Fig. 2A: Please correct in the displayed sequence alignment the name "Anabaene" to "Anabaena" and add to the legend to which Anabaena species or strain this sequence belongs to, as you did for all the other sequences.

Reviewer #1 (Remarks to the Author):

The author has clarified the queries raised and also introduced certain sentences for better clarity. The manuscript reads well and gives a good information on the repressor.

The work should be published

Reviewer #2 (Remarks to the Author):

This work provides substantial conceptual advance and has been carefully conducted. During the revision, all my previously raised points have been addressed very well and the paper should be published.

I just have the following two minor points that still need correction:

L392: Please correct spelling of "Prochorococcus" to "Prochlorococcus".

Fig. 2A: Please correct in the displayed sequence alignment the name "Anabaene" to "Anabaena" and add to the legend to which *Anabaena* species or strain this sequence belongs to, as you did for all the other sequences.

Thanks for the comments of the reviewers. We corrected these mistakes. In line 392, we corrected the spelling of "Prochorococcus" to "Prochlorococcus". In Figure 2A, we corrected the name "Anabaene" to "Anabaena" and added the information " *Anabaena*: *Anabaena* sp. PCC 7120;" into the legend.